# Rectangular Amplitude Mask-Based Auto-Focus Method with a Large Range and High Precision for a Micro-LED Wafer Defects Detection System

**DOI:** 10.3390/s23177579

**Published:** 2023-08-31

**Authors:** Wenjun He, Yufeng Ma, Wenbo Wang

**Affiliations:** College of Opto-Electronic Engineering, Changchun University of Science and Technology, Changchun 130022, China; mayf1234562023@163.com (Y.M.); wangwb202303@163.com (W.W.)

**Keywords:** Micro-LED, auto-focus, eccentric beam, rectangular amplitude mask

## Abstract

Auto-focus technology plays an important role in the Micro-LED wafer defects detection system. How to accurately measure the defocus amount and the defocus direction of the Micro-LED wafer sample in a large linear range is one of the keys to realizing wafer defects detection. In this paper, a large range and high-precision auto-focus method based on a rectangular amplitude mask is proposed. A rectangular amplitude mask without a long edge is used to modulate the shape of the incident laser beams so that the spot shape distribution of the reflected laser beam on the sensor changes with the defocus amount of the wafer sample. By calculating the shape of the light spots, the defocus amount and the defocus direction can be obtained at the same time. The experimental results show that under the 20× microscopy objective, the linear range of the auto-focus system is 480 μm and the accuracy can reach 1 μm. It can be seen that the automatic focusing method proposed in this paper has the advantages of large linear range, high accuracy, and compact structure, which can meet the requirements of the Micro-LED wafer defects detection equipment.

## 1. Introduction

The Micro-LED (light-emitting diode) display is a new generation of display devices that can integrate drivers, light emitting, and signal transmission, and it can realize an ultra-large-scale integrated light-emitting unit. With some excellent characteristics, such as small chip size, high integration, self-emitting, and low energy consumption [1,2], the Micro-LED display has great advantages in brightness, resolution, contrast, color gamut, and energy consumption compared with LCD (liquid crystal display) and OLED (organic light-emitting diode) in display [3,4,5]. However, the mass production of Micro-LED displays still faces some problems such as miniature chips and low yield. In the chip manufacturing process, how to quickly and accurately detect defective chips automatically is the key to improving the yield rate, and it is also one of the key technologies for Micro-LED wafers to achieve industrialization in the future [6,7]. An infinite conjugate microscope is usually used to optically magnify the tiny chip patterns in the Micro-LED wafer defect detection optical system. Due to the short depth of focus of the microscope, it is extremely sensitive to defocus. Once the wafer surface deviates from the focus plane, the image quality is rapidly reduced, which will seriously affect the detection accuracy of the chip and increase the rate of missed detection or false detection. Therefore, the auto-focusing technology is essential in the Micro-LED wafer defects detection optical system. To achieve the accurate, fast and efficient detection of Micro-LED wafer defects, auto-focus technology should have some remarkable features, such as a large range of defocusing detection, high precision and short response time of auto-focusing.

At present, auto-focus technology can be divided into two categories: passive auto-focus (PAF) [8,9,10,11,12,13] and active auto-focus (AAF) [14,15,16,17,18,19,20,21]. The PAFs mainly rely on image processing algorithms to evaluate the image definition and predict the defocus amount by referring to the definition value, which does not require additional light sources and auxiliary optical paths. In recent years, many scholars have proposed a variety of clarity evaluation algorithms and search algorithms to improve the performance of the PAF. In 2021, D. Peter proposed a search algorithm based on the Gaussian standard deviation and gradient block [12]. In 2022, C. Zhang proposed an auto-focus method based on a multi-interesting area window, which applied multiple medium-value filtering and the accuracy of the right to noise to improve the noise improvement [13]. Due to the advantage of low cost, PAFs are widely used in many industries. However, the analysis of the image is complex and too many iterations are needed, which makes it hard to meet the precision and real-time requirements for the Micro-LED wafer defects detection [12,13]. In contrast, the AAFs require additional light sources (mostly a laser) and specially designed optical paths. The AAFs mainly rely on the linear relationship between the defocus amount and the light spot information (intensity or spot pattern) on sensors to achieve auto-focus. In 2018, G. Chen proposed a method to obtain the defocus amount by offsetting the image plane and calculating the light spot radius based on the clustered circular alignment algorithm [14]. In 2019, C. Liu proposed a laser-based auto-focusing microscope for a sample with a transparent boundary layer, a polarized beam-splitting system, and an image processing program combined in this microscope [15]. The AAFs are usually characterized by a high cost and a more complex structure given because of the additional hardware [14,18,20]. Nevertheless, the AAFs are found to be more suitable for production line applications such as Micro-LED wafer defects detection than the PAFs because the AAFs can provide fast focusing in a short response time with high accuracy, and they are not sensitive to the content in the field of view of the microscope [21].

For different types of Micro-LED wafer defects, the inspection process usually requires the replacement of microscopes with different optical magnifications, which also have different focal depths. Therefore, the auto-focus module needs to have a large linear range and high accuracy. In this paper, we propose a large range and high-precision auto-focus method based on a rectangular amplitude mask for a Micro-LED wafer defects detection system. A rectangular amplitude mask without a long edge is used to modulate the shape of the incident laser beams so that the position and spot shape distribution of the reflected laser beam on the sensor change with the defocus amount of the wafer sample. In addition, the reflected laser beam has different centroid change rates on the *x*-axis section and *y*-axis section of the sensor. The *x*-axis with a low centroid change rate can achieve coarse auto-focus within a large linear range, while the *y*-axis with a high centroid change rate can achieve fine auto-focus within a small linear range. The combination of these two methods is effective and can achieve accurate automatic focusing within a large linear range. This method solves the problem of compatibility between a large linear range and high focusing precision. And using the linear relationship between the light spots and the defocus amount can quickly and accurately calculate the defocus information to ensure real-time focusing. The remaining content of this paper is arranged as follows: Section 2 provides a theoretical derivation of the proposed auto-focus method, Section 3 provides simulation and experimental results of this method, and Section 4 summarizes the work of this paper.

## 2. Methods

The composition of the automatic focusing system based on a rectangular amplitude mask is shown in Figure 1. A linearly polarized laser beam with a wavelength of 650 nm is collimated and expanded by a beam expander. The cross-sectional shape of the laser beam is adjusted to a rectangular ring without a long edge by a rectangular amplitude mask. A half-wave plate (HWP) is used to adjust the polarization direction of the linearly polarized laser beam to coincide with the characteristic direction (i.e., *s*-polarized) of the polarization beam splitter (PBS), so that the laser beam can pass through the PBS without reflection energy loss. After passing through a quarter-wave plate (QWP), the polarization state of the laser beam is changed from *s*-polarized to right-handed circularly polarized. Then, the laser beam is focused on the wafer sample by the microscopic objective (infinite conjugation) and reflected by the sample surface. Due to the opposite propagation direction, the polarization state of the reflected laser beam becomes left-handed circularly polarized, and then it passes through the QWP again and is modulated into *p*-polarized. The *p*-polarized laser beam is completely reflected by the PBS and received by the sensor (CCD or CMOS) after passing through the tube lens. The light spot signal (i.e., position and spot shape distribution) of the reflected laser beam is linearly related to the defocus direction and defocus amount of the wafer sample. At the same time, the infinite conjugate microscopic objective is fixed on a precision displacement stage that can move rapidly along the optical axis, forming a feedback system with the light spot signal on the sensor to achieve automatic focusing.

This method relies on a rectangular amplitude mask to modulate the cross-section of the incident laser beam into a rectangular ring (missing a long edge) so that the shape information of the light spot includes the defocus information of the wafer sample, which includes three situations: inner-focus, in-focus, and outer-focus. And the corresponding spot shapes are a rectangular ring with a left opening, a dot, and a rectangular ring with a right opening, respectively, as shown in Figure 1. When the wafer sample moves from the inner-focus position to the outer-focus position, the shape of the laser beam received by the sensor will gradually change from the rectangular ring with a left opening to the rectangular ring with a right opening, and the length of the rectangular rings will decrease first and then increase. Therefore, the defocus direction of the wafer sample can be determined by the opening direction of the rectangular ring, and the defocus amount can be calculated by the length of the rectangular ring.

The schematic diagram of the auto-focus principle is shown in Figure 2. According to the principle of geometrical optics, the spot position $\Delta_{i}$ on the sensor has the following relationship with the defocus amount δ [21]: (1)$$\Delta_{i}=\frac{2d_{i}f_{1}}{f_{0}^{2}}\delta ,$$
where $d_{i}$ is the radius of the incident laser beam, $f_{0}$ is the focal length of the infinite conjugation microscope objective, and $f_{1}$ is the focal length of the tube lens.

To rigorously quantify this relationship, the center of the light spot is defined as ($i_{c}$,$j_{c}$) when the wafer sample is in the in-focus position, and then the xy Cartesian coordinate system is established with the coordinate origin of ($i_{c}$,$j_{c}$), as shown in Figure 3a. In Figure 3b, the energy center of the light spot (i.e., the *x*-axis image centroid $i_{cog}$) changes with the wafer sample position, so the relationship between the image centroid value $i_{cog}$ and the defocus amount can be constructed. The defocus amount is calculated based on the $i_{cog}$ value, where the positive and negative $i_{cog}$ values represent the defocus direction.

In summary, the defocus amount can be calculated using the energy center of the light spots on the *x*-axis cross-section (corresponding to the centroid of the *x*-axis image on the sensor). Figure 4 shows a schematic diagram of the light spot on the sensors when the wafer sample is in the inner-focus or outer-focus position, and the calculation method of the centroid of the *x*-axis image is shown as [21]: (2)$$i_{cog}=\frac{\int_{\Delta_{1}}^{\Delta_{2}}\left(\Delta_{i}\times p_{x}\left(i\right)\right)}{\int p_{x}\left(i\right)},$$
where $\Delta_{1}$ and $\Delta_{2}$ are 1/2 of the inner side length and 1/2 of the outer side length of the short side of the rectangular ring on the sensor, respectively. And $p_{x}\left(i\right)$ represents the intensity (i.e., gray value) located at the coordinate *i* on the *x*-axis. Combining Formulas (1) and (2), $i_{cog}$ can be expressed as,
(3)$$i_{cog}=\frac{2f_{1}\int_{d_{1}}^{d_{2}}\left(d_{i}\times p_{x}\left(i\right)\right)}{f_{0}^{2}\int p_{x}\left(i\right)}\delta ,$$
where $d_{1}$ and $d_{2}$ are 1/2 of the inner edge length and 1/2 of the outer edge length of the short side of the rectangular amplitude mask, respectively. From Formula (3), it can be obtained that there is a linear relationship between the centroid $i_{cog}$ of the *x*-axis image and the defocus amount δ, so automatic focusing can be achieved by using this linear relationship.

As shown in Figure 3c, higher detection sensitivity and accuracy can be obtained when we analyze the *y*-axis cross-section of the light spot on the sensor. The *y*-axis cross-section has two energy centers (corresponding to two centroid positions), denoted as $j_{+cog}$ and $j_{-cog}$, respectively. The defocus amount can be calculated using the difference $j_{TC}$ between these two centroid positions.
(4)$$j_{TC}=j_{+cog}-j_{-cog},$$
from Formula (3), it can be obtained that
(5)$$j_{+cog}=\frac{2f_{1}\int_{d_{3}}^{d_{4}}\left(d_{i}\times p_{x}\left(i\right)\right)}{f_{0}^{2}\int p_{x}\left(i\right)}\delta ,$$
and
(6)$$j_{+cog}=-j_{-cog},$$
bringing Formulas (5) and (6) into Formula (4) yields: (7)$$j_{TC}=2j_{+cog},$$
where $d_{3}$ and $d_{4}$ are 1/2 of the inner edge and 1/2 of the outer edge of the long side of the rectangular amplitude mask, respectively. Since the size of the long side of the rectangular amplitude mask is designed to be larger than the size of the short side, the parameters of the image on the sensor in Figure 4 satisfy the following equation,
(8)$$\left|j_{+cog}\right|>\left|i_{cog}\right|,$$

Hence, we can obtain
(9)$$\left|j_{TC}\right|>\left|2i_{cog}\right|,$$

It shows that the sensitivity and accuracy of $j_{TC}$ are more than twice that of $i_{cog}$, and the specific values are determined by the sizes of the long and short sides of the rectangular amplitude mask. However, the $j_{TC}$ has the same sign whether the wafer sample is in the inner-focus position or the outer-focus position, so we cannot directly determine the defocus direction by calculating the $j_{TC}$. Compared with $i_{cog}$, $j_{TC}$ brings higher accuracy, but the corresponding linear region is reduced. Combining these two methods can achieve high-precision auto-focus over a larger range, i.e., the positive or negative sign of the $i_{cog}$ is used to determine the defocus direction, and the value of $j_{TC}$ is used to calculate the defocus amount δ.

## 3. Results and Discussion

### 3.1. Simulation Analysis

To obtain reasonable and optimal optical parameters, an optical simulation model of the automatic focusing system is established by using VirtualLab Fusion 7.6.1.18 software. In this simulation model, the infinite conjugate objective lens in Figure 1 is replaced by an ideal lens with a focal length $f_{0}$ of 40 mm and a diameter of 15 mm, which simulates a microscope objective with a magnification of 5 while avoiding the adverse effects of optical aberrations. The focal length of tube lens $f_{1}$ is equal to 200 mm. The short side of the rectangular amplitude mask is 4 mm inside and 6 mm outside; the long side is 6 mm inside and 8 mm outside. The sensor has 2592 × 1944 pixels with a pixel size of 2.2 μm, so the size of the image plane is 5.7 mm × 4.27 mm. The linearly polarized laser beam is simulated by a Gaussian wave with a waist radius of 5 mm and a radius of 10 mm. And the wafer sample is replaced by a mirror. When the mirror is moved quantitatively near the focus plane of the infinite conjugate objective lens, different light spot shapes and positions on the image plane can be obtained. Then, the data of the light spot information are processed by Matlab 2020a [22], and the relationship between the defocus amount and centroid value is obtained, as shown in Figure 5.

In Figure 5, both $i_{cog}$ and $j_{TC}$ show straight linear relationships with the defocus amount. The linear region of the $j_{TC}$ fitting result is smaller than the $i_{cog}$ fitting result, but the slope of the y-axis centroid $j_{TC}$ fitting is more than twice that of the x-axis centroid $i_{cog}$ fitting. The slope of the fitting line represents the defocusing sensitivity of the automatic defocusing system, which indicates that the sensitivity of the y-axis centroid $j_{TC}$ fitting result is more than twice that of the x-axis centroid $i_{cog}$ fitting result. In addition, when the mirror is in the inner-focus and outer-focus positions, $i_{cog}$ has opposite signs, while $j_{TC}$ has the same sign. Therefore, the positive and negative of $i_{cog}$ can be used to predict the defocus direction, and the value of $j_{TC}$ can be used to calculate the defocus amount. Therefore, the simulation results are consistent with the theory in Section 2.

If we only change the focal length of the infinite conjugate objective lens and keep the other parameters in the optical simulation model unchanged, we can obtain the simulation results for microscope objectives with different magnifications. Figure 6 shows the simulation results of a 20× microscopic objective, and its focal length is equal to 10 mm. Comparing Figure 5 and Figure 6, it can be seen that when the focal length $f_{0}$ of the microscope objective decreases (corresponding to an increase in magnification), the defocus sensitivity of $i_{cog}$ and $j_{TC}$ both improve, but the corresponding linear regions decrease.

Then, we changed the parameters of the rectangular amplitude mask to the short side of 2 mm inside and 4 mm outside, and the long side of 4 mm inside and 6 mm outside, ensuring that the focal length of the microscope objective remains equal to 40 mm. The simulation results are shown in Figure 7, where it can be observed that as the size of the rectangular amplitude mask decreases, the defocus sensitivity of $i_{cog}$ and $j_{TC}$ both decrease, but the corresponding linear regions increase. It should be noted that the defocus sensitivity can theoretically be improved by increasing the size of the rectangular amplitude mask, but it is limited by the entrance pupil diameter of the microscope objective in practical applications.

As described in Section 2, $j_{TC}$ is used to calculate the defocus amount and the sign of $i_{cog}$ is used to determine the defocus direction, so that the automatic focusing system can achieve a high-precision and large linear range at the same time. This means that the short side of the rectangular mask can theoretically be very short, because the slope of the $i_{cog}$ is not used in the data proceeding. However, if the Fraunhoff diffraction of the rectangular aperture is considered, a very small size of the short side will produce a significant diffraction spot when the wafer sample is near the focus plane, resulting in a wrong interpretation of the defocus direction. We comprehensively consider the linear region, defocus sensitivity, and the diffraction effect, and the final parameters of the rectangular amplitude mask are designed as a short edge of 2 mm inside and 3 mm outside, a long edge of 4 mm inside and 5 mm outside.

### 3.2. Experiment and Discussion

According to the composition diagram of the automatic focusing system shown in Figure 1, an experimental optical path is built for the experiment, as shown in Figure 8. In the experiment, a semiconductor laser with a wavelength of 650 nm (RealLight, AWS 650-ISF-002) is used as the light source, and it cooperates with a fiber collimator (RealLight, RL-CL51) to illuminate the rectangular amplitude mask (chrome-plated glass plate, short side: 2 mm inside and 3 mm outside, long side: 4 mm inside and 5 mm outside). A diaphragm (Daheng Optics, GCM-5701M) is used to adjust the size of the laser beam. Polarization elements such as a HWP (Daheng Optics, GCL-060652), a PBS (THORLABS, CCM1-PBS252/M), and a QWP(Daheng Optics, GCL-060642) are used to change the polarization states of the laser beam, thereby improving light energy utilization and reducing stray light. The model of the microscopic objective is the M plan Apo of Mitutoyo company, with a numerical aperture of 0.42, a depth of focus of 1.6 µm, and a magnification of 20×. To accurately evaluate the measurement error of the defocus amount, the wafer sample is replaced by a mirror (Daheng Optics, GCC-102102). A tube lens (Daheng Optics, GCL-010146, focal length f = 150 mm) and a CCD camera (Daheng Optics, MER-500-7UM) are used to capture the images of the laser beam. In addition, a dual-frequency laser interferometer (Renishaw, laser XL-80, measurement resolution of 1 nm) is used to accurately measure the displacement of the mirror as the true value of the defocus amount.

#### 3.2.1. Verification of the Linear Relationship

When the mirror moves from the inner-focus position to the outer-focus position, the trend of changes in the shape of the light spot collected by the CCD is shown in Figure 9. It can be seen that when the mirror is in the inner-focus position, the spot is a rectangular ring opening to the left. As the defocus gradually decreases, the size of the rectangular ring also decreases. When it reaches the in-focus position, the rectangular ring becomes a spot. But due to the diffraction effects, some diffraction spots appear around the central spot in both the *x*-axis and *y*-axis. When the mirror moves to the outer-focus position, the central spot also becomes a rectangular ring opening to the right. And as the defocus amount increases, the size of the rectangular ring also increases.

Due to the influence of the experimental environment, we have to use some simple image processing algorithms [23]. Firstly, the collected images are filtered by the median filtering algorithms to reduce the influence of diffraction, and then image noise points are removed by the OTSU method [24,25,26]. The obtained image centroid data and linear fitting relationship are shown in Figure 10. In Figure 10a, it can be seen that for the *x*-axis section, the linearity of the fitting of the spot centroid data $i_{cog}$ with the defocus amount δ is good, with a linear range of 800 μm. As shown in Figure 10b, the linearity of the fitting of the *y*-axis centroid data with the defocus amount is also very good, with a linear range of 480 μm. The fitted line is divided into two segments by the focal plane. The slope of each straight line is more than twice that of the $i_{cog}$ straight line, so it is possible to use the $i_{cog}$ straight line for coarse autofocus in a large linear range and the $j_{TC}$ straight line for fine autofocus in a small linear range. The combination of the two achieves high-precision autofocus in a large linear range, and the experimental phenomenon is consistent with both theory and simulation results. During the experimental process, due to installation and adjustment reasons, it is difficult to ensure that the centers of each experimental device are strictly on the optical axis, which will have a small impact on the linearity of the fitting line. Moreover, when the mirror is close to the focal plane, the diffraction effect is obvious, which will have a significant impact on the linearity of the fitting line near the focal plane, thereby affecting the focusing accuracy.

#### 3.2.2. Measurement Error of the Defocus Amount

The measurement error of the defocus amount was tested by incorporating the centroid values obtained from the experiment into the fitted linear equation and then comparing them with the true values of the defocus amount, which were obtained by the dual-frequency laser interferometer. Firstly, we evaluate the influence of the diffraction effect on the measurement error of the defocus amount. When the image processing algorithms are not used to preprocess the spot image, the measurement error data of the $j_{TC}$ fitting line are obtained as follows: (1) when the defocus amount is equal to zero, the measurement error is 10.3 μm; (2) when the defocus amount is equal to +0.02 mm and −0.02 mm, the measurement errors are 7 µm and 9.7 μm, respectively; (3) when the defocus amount is equal to +0.04 mm and −0.04 mm, the measurement errors are 4 μm and 2.3 μm, respectively. Secondly, we use the median filtering and the OTSU algorithm to preprocess the spot image, and the test results are shown in Figure 11. The diffraction effect is obviously suppressed as follows: (1) when the defocus amount is equal to zero, the measurement error is 3.7 μm; (2) when the defocus amount is equal to +0.02 mm and −0.02 mm, the measurement errors are 4 μm and 3.8 μm, respectively; (3) when the defocus amount is equal to +0.04 mm and −0.04 mm, the measurement errors are 2.9 μm and 1.2 μm, respectively. In Figure 11, it can be seen that the error values of $j_{TC}$ are all less than $i_{cog}$, which again corresponds to the theory. Outside the ±0.06 mm range of the focal plane, the error values of $j_{TC}$ are all less than 1 μm, which are also less than the depth of focus of the 20× microscope objective. However, within the ±0.06 mm range of the focal plane, there is a significant error due to diffraction phenomena. To solve this problem, we propose the following solution: (1) when the defocus amount is equal to 0.06 mm, the value of $j_{TC}$ is marked as $j_{0}$; (2) if the $j_{TC}$ value of the collected image is less than $j_{0}$, the wafer sample will be moved 0.06 mm far away from the focal plane; (3) then we collect the image again, calculate the spot information, and complete automatic focusing based on the newly collected image. This solution ensures that the automatic focusing system has a focusing accuracy of better than 1 μm over a linear detection range of 480 μm. Similarly, it is easy to obtain a focusing accuracy of 2 μm over a linear detection range of 800 μm.

## 4. Conclusions

In this paper, an automatic focusing method based on a rectangular amplitude mask is proposed, which improves the traditional eccentric beam method by using a rectangular amplitude mask to modulate the incident laser beam to a rectangular ring cross-section (missing a long edge). A rectangular ring spot is formed on the sensor, and then, the long edge (*x*-axis) of the rectangular ring with a small rate of change in the center of mass is used to perform coarse focusing over a large range. Utilizing the short edge (*y*-axis) of a rectangular ring with a large rate of change in the center of mass for precise focusing in a small range enables high-precision automatic focusing in a large range. The experimental results show that this method can achieve automatic focusing with an accuracy of 1 μm over the linear range of 480 μm, and the measurement error is less than the depth of focus of the microscope objective. This work provides a new method for the field of automatic focusing, which can meet the requirements of Micro-LED wafer defects detection equipment.

## Figures and Tables

**Figure 1 sensors-23-07579-f001:**
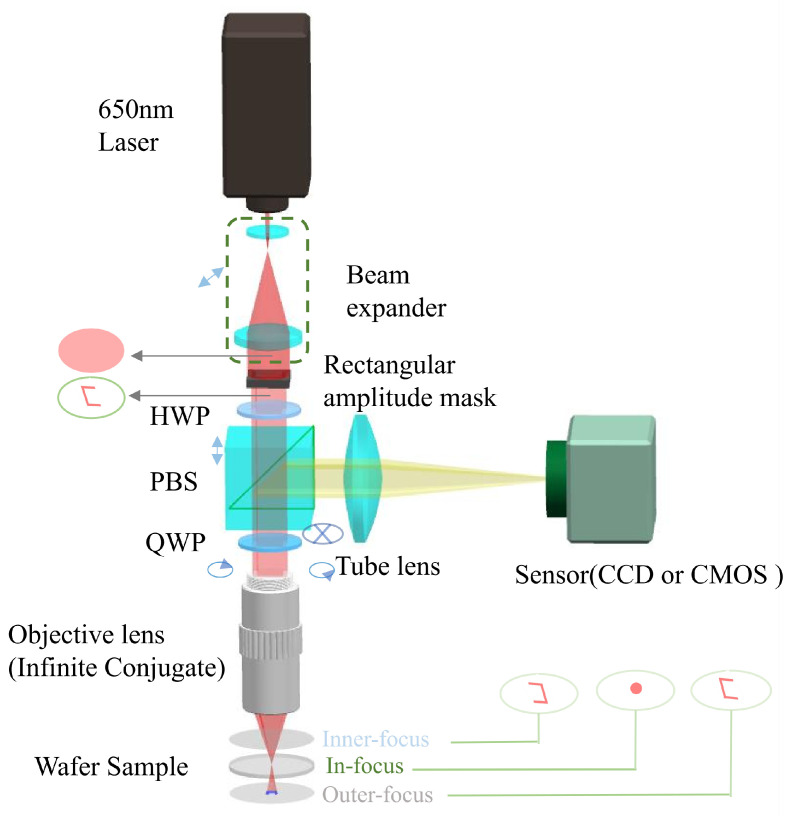
Composition diagram of the automatic focusing system.

**Figure 2 sensors-23-07579-f002:**
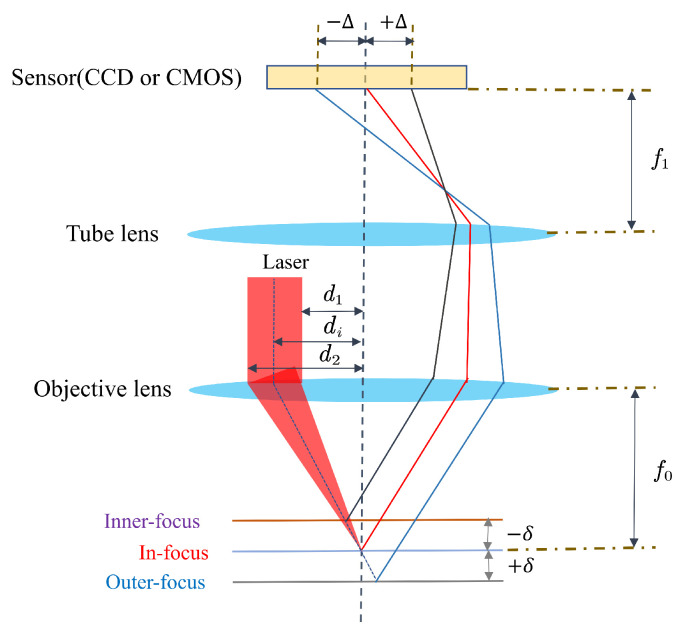
Schematic diagram of the auto-focus principle.

**Figure 3 sensors-23-07579-f003:**
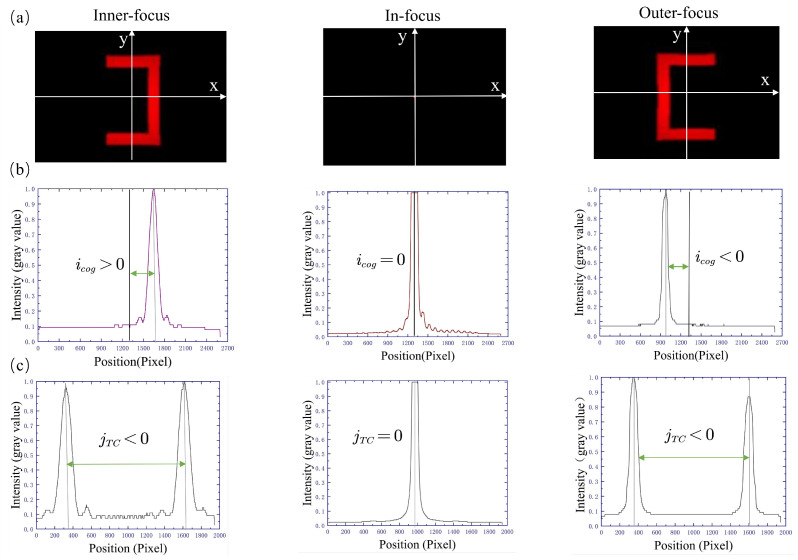
(**a**) Shape of light spot images with different defocus situations; (**b**) *x*-axis centroid schematic diagram; (**c**) *y*-axis centroid schematic diagram.

**Figure 4 sensors-23-07579-f004:**
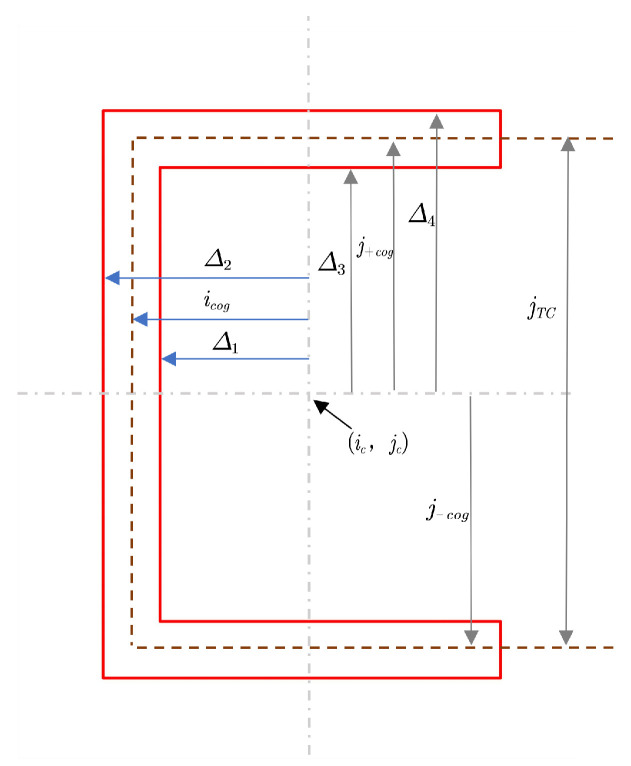
Schematic diagram of the light spot when the wafer sample is in the defocus position.

**Figure 5 sensors-23-07579-f005:**
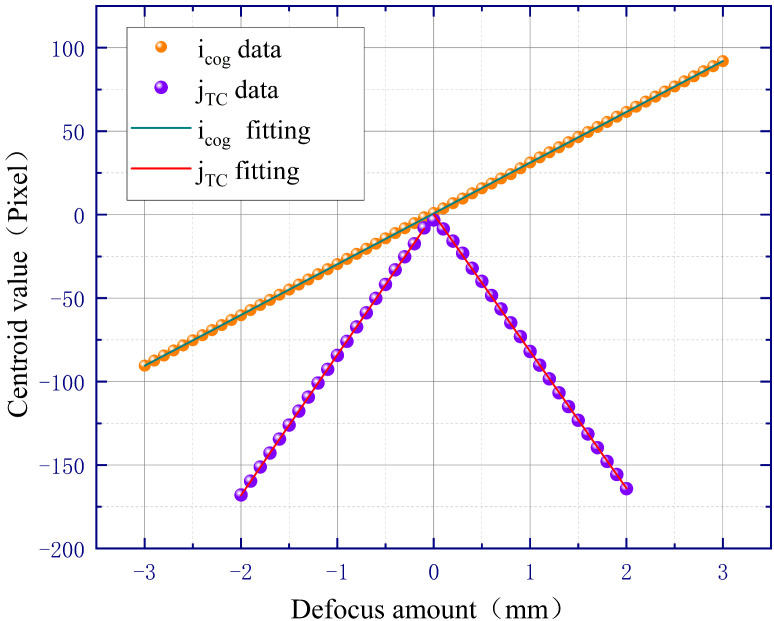
Simulation results of 5× microscopic objective.

**Figure 6 sensors-23-07579-f006:**
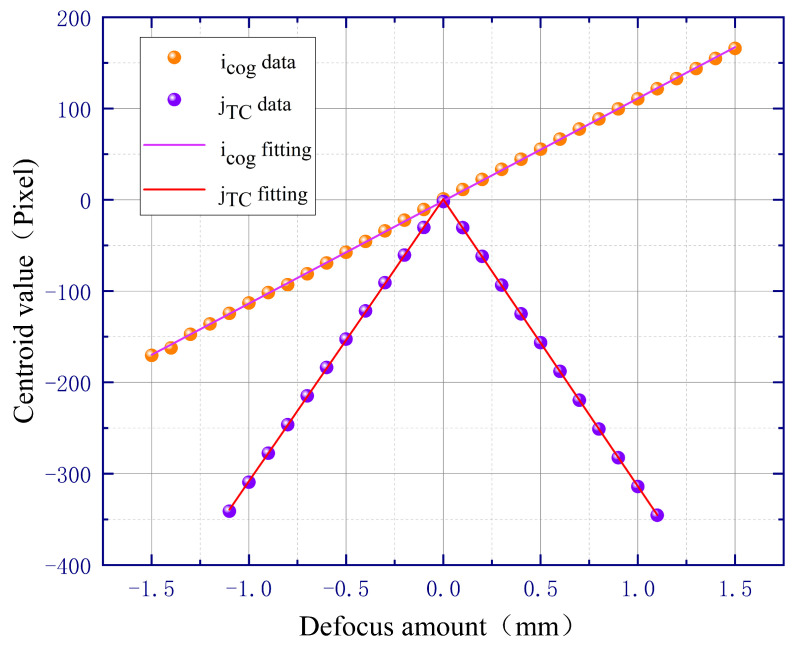
Simulation results of 20× microscopic objective.

**Figure 7 sensors-23-07579-f007:**
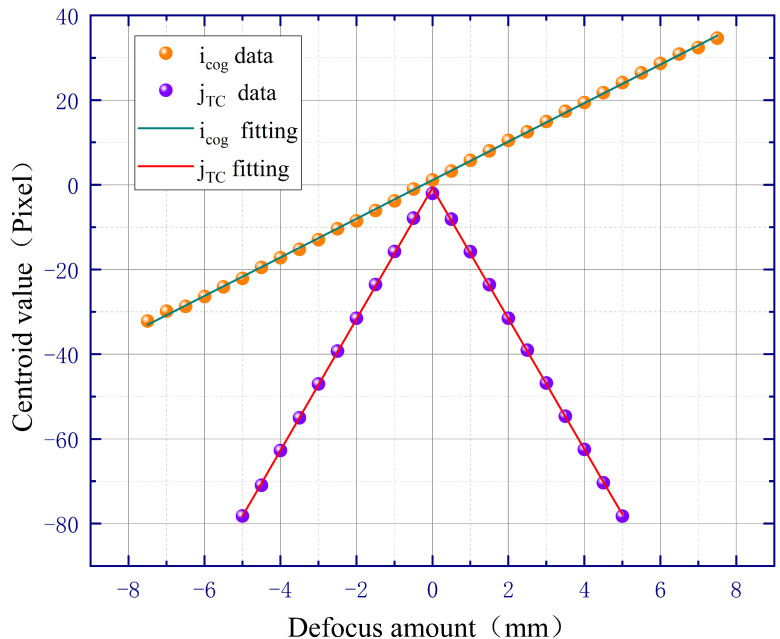
Simulation results of 5× microscopic objective after changing amplitude mask parameters.

**Figure 8 sensors-23-07579-f008:**
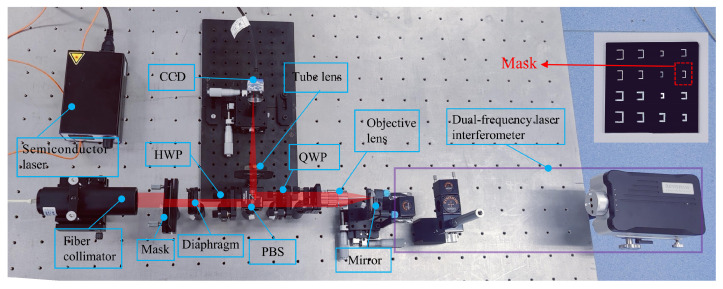
Photo of the experiment platform.

**Figure 9 sensors-23-07579-f009:**
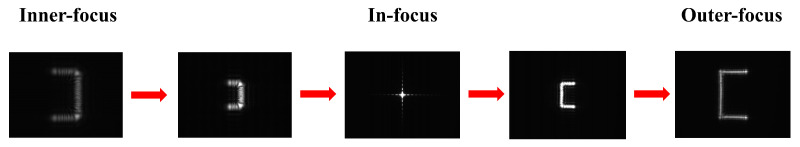
Change trend chart of the light spot on the CCD.

**Figure 10 sensors-23-07579-f010:**
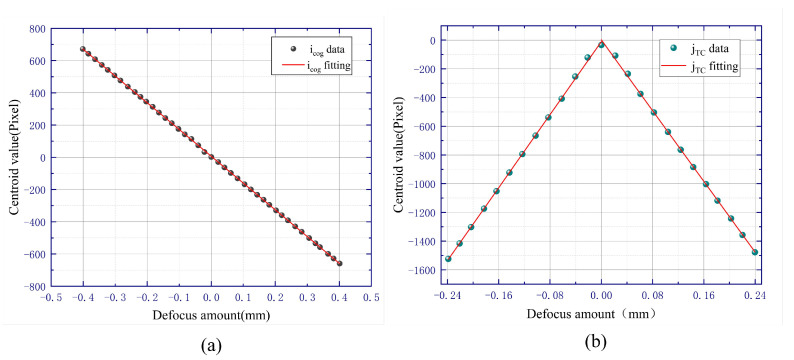
Linear relationship diagram: (**a**) relationship between centroid value $i_{cog}$ of the *x*-axis and defocus amount. (**b**) relationship between centroid value $j_{TC}$ of the *y*-axis and defocus amount.

**Figure 11 sensors-23-07579-f011:**
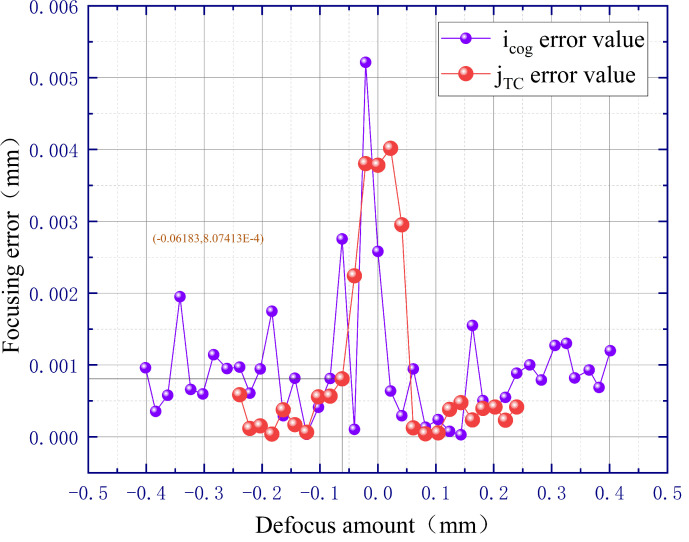
Measurement error of the defocus amount.

## Data Availability

Not applicable.

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
