# Peer review of "Rectangular Amplitude Mask-Based Auto-Focus Method with a Large Range and High Precision for a Micro-LED Wafer Defects Detection System"

_sensors, 2023, doi:10.3390/s23177579_

Round 1
Reviewer 1 Report
This manuscript systematically explains an auto-focus method for a Micro LED wafer defects detection system with rectangular amplitude masks based auto-focus method. The work is rigorous and complete, including its theoretical derivation simulation and experimental proof. The new and innovative technology first time use a rectangular amplitude mask to modulate the laser beam cross-section, which is of great significance for improving the accuracy and linear range of autofocus technology.
The manuscript is well written. I would recommend this paper for publication in Sensor after revision. Some detailed comments are appended below.
1) In the manuscript, two imaging algorithms, median filtering and OTSU, are specifically used to process the light spot. Can we achieve higher accuracy by optimizing the image processing algorithm?
2) In the experiment, HWP and QWP are used to modulate the polarization state of the incident light, which results in more optical components for the system. Does this have practical significance?
3) Formula 5 and Formula 6 are incorrect. It recommends to append absolute value signs.
4) In Section 3.1, in the simulation analysis, it recommends to append concerning parameters for the simulated light source.
Reviewer 2 Report
The submitted manuscript is entitled: Rectangular amplitude mask based auto-focus method with a large range and high precision for a Micro LED wafer defects detection system.
Overall, this work provides a promising strategy for the field of automatic focusing, which can meet the requirements of micro-LED wafer defects detection equipment. The findings of this study could be interesting to the readership of the Sensors journal.
The manuscript should be revised considering the following minor issues:
The use of some generalizations seems imprecise. To be precise, you should avoid the phrase ‘micro LED’ without specifying. The 'micro-LED technology/ display' etc. seems appropriate.
Line 215: CCD is a sensor; this should read: a CCD camera.
The section names should be reorganized. Traditional sections e.g. Materials and Methods and Results and Discussion seem more legible for the readers.
Figure 5 shows the simulation results for the 5x objective. However, in the text, this magnification is not discussed.
Line 245: Can the authors estimate the impact of diffraction effects on the focusing accuracy?
Line 252: The sentence should read: The test results are shown in Figure 11.
The space between values and units should be added.
In some cases, the use of the symbol ± seems inappropriate.
The micrometer unit is not correct.
Reviewer 3 Report
This article proposes a large range and high-precision auto-focus method based on a rectangular amplitude mask. Some shortcomings of the article and several suggestions are shown below:
a) Among the most critical metrics in the performance validation of micro LEDs are the Time-resolved photoluminescence spectra (TRPL) for LEDs with different quantum well (QW) thicknesses, external quantum efficiency (EQE) for LEDs with different QW widths, and 3 -dB modulation bandwidths vs current density. No validation is observed in the work under any of these metrics.
b) The introduction must be improved in its final part. The contributions of the article must be explicit in bullets or literal for a correct understanding.
c) The abbreviations are poor. Many abbreviations that are in the document are missing in the abbreviations section
There are many types throughout the document. A thorough review of English is recommended.
Reviewer 4 Report
The manuscript entitled ”Rectangular amplitude mask based auto-focus method with a large range and high precision for a Micro LED wafer defects detection system.“ Wenjun He et al. reports focusing method based on a rectangular amplitude mask is proposed for the Micro-LED wafer defects detection system.. The research work is interesting, and this study of the edge extraction method is much needed in the Micro LED Defect detection optical system. The authors have performed quite good research on this investigation. However, the article needs some clarification. The language used in this article is reasonable. The critical statement to support the results/finding and the statement's citation must be included. Overall, the article is worthy. However, all the results are needed to connect, and a thorough description by providing detailed information can improve the fineness of this manuscript. I want to address a few queries on this manuscript, which will help improve the quality of the article. Please find the comment below.
1. Page no 1, line 9, 240 um, does the u stands for micron (µ)
2. Page no 1, line 14, “Micro LED is a new generation of display devices that can integrate drivers, light emitting, and signal transmission, and it can realize an ultra-large-scale integrated light- 16emitting unit “ [**].Please provide suitable references if you have any.
3. Page no 1, line 20, However, the mass production of Micro LED still faces some problems such as miniature chips and low yield [**]. Please provide suitable references if you have any.
4. Page no 2, line 43, However, the analysis of the image is complex and too many iterations are needed, which makes it hard to meet the precision and real-time requirements for the Micro LED wafer defects detection [**].Please provide suitable references if you have any.
5. Page no 2, line 55, Nevertheless, the AAFs are found to be more suitable for production line applications such as Micro LED wafer defects detection than the PAFs[**].Please provide suitable references if you have any.
6. Page no 2, line 56, Because the AAFs can provide a fast focusing in a short response time with high accuracy, and they are not sensitive to the content in the field of view of the microscope[**].Please provide suitable references if you have any.
7. Page no 2, line 58, “For different types of Micro LED wafer defects, the inspection process usually requires the replacement of microscopes with different optical magnifications, which also have 6different focal depths. Therefore, the auto-focus module needs to have a large linear range and high accuracy [**].Please provide suitable references if you have any.
- On page 2, line 82, the Author state,” After passing through a quarter-wave plate (QWP), the polarization state of the laser beam is changed from s-polarized to right-handed circularly polarized”. The linearly polarized state, having the two orthogonal components, makes circularly polarized light after passing through QWP. Here S, polarized light has no orthogonal components. How it makes the right circularly polarized light after passing through QWP?
- In Figure 1. Composition diagram of the automatic focusing system.[*]. It is a prevalent interferometer system for displacement measurement. I suggest giving some standard references so that researchers can understand how this interferometer works through the application here is different.
- Page no 3, line 107, According to the principle of geometrical optics, the spot position Δi on the sensor has the following relationship with the defocus amount δ [**]. Please provide suitable references if you have any.
- In Figure 2, What is the distance between the + δ and – δ? It would be good to mark the values of f0 and f1 used in the investigation.
- In Figures 3(b), and 3(c), font clarity needs to improve.
13. Page no 4, line 121, Figure 4 shows a schematic diagram of the light spot on the sensors when the wafer sample is in the Inner-focus or Outer-focus position, and the calculation method of the centroid of the x-axis image is shown as [*].Please provide a suitable references if you have any.
- Do the authors derive equations (3) and 4? I suggest providing the detailed derivation in a separate appendix section.
- In Figure 9, the inner and outer focus images are not the same as those represented in Figure 3 (a). Why.
16. Page no 9, line 229, Due to the influence of the experimental environment, we have to use some simple image processing algorithms [**].Please provide a suitable references if you have any.
17. Page no 9, line 230, Firstly, the collected images are filtered by the median filtering algorithms to reduce the influence of diffraction, and then image noise points are removed by the OTSU method [**]. Please provide a suitable references if you have any.
- In Figure 10, the x-axis de-focusing amount is linearly decreasing, whereas in the y-axis, it is different; the centroid value is also different from the x-axis. Why this discrepancy?
- Figure 11, How do authors estimate the focusing error?
- What does the author mean by auto-focus in this investigative work?
Round 2
Reviewer 3 Report
The article has improved substantially. However, many acronyms are still missing from the acronym table to give the manuscript a serious and coherent structure.
There are still grammar errors, for example in the introduction.
Reviewer 4 Report
The revised manuscript entitled “Rectangular amplitude mask based auto-focus method with a large range and high precision for a Micro LED wafer defects detection system.“ Wenjun He et al. report focusing technique based on a rectangular amplitude mask is proposed for the Micro-LED wafer defects detection system. The authors have significantly improved the review manuscript by illustrating their research regarding modifications, figures and contents. Also, the authors have given satisfactory responses to the comment raised. The article will be the appropriate form for publication in sensors.
